# Structural Dynamics in the Presence of Water of Graphene Bilayers with Defects

**DOI:** 10.3390/nano13142038

**Published:** 2023-07-10

**Authors:** Elizabeth Santos

**Affiliations:** Institute of Theoretical Chemistry, Ulm University, Mez-Starck-Haus, Oberberghof 7, 89081 Ulm, Germany; esantos@uni-ulm.de

**Keywords:** graphene, vacancies, Jahn–Teller distortion, quantum mechanical molecular dynamics, tight binding

## Abstract

The dynamics of a bilayer of graphene containing one mono-vacancy in the top layer has been investigated in the framework of DFTB in the absence and in the presence of water. Due to the speed of the code, we can describe details of the behavior, which are not directly accessible experimentally and cannot be treated by DFT or classical molecular dynamics. The presence of water enhances the displacement of carbon atoms in the perpendicular direction to the surface. Our results explain very well a variety of experimental findings. In particular, the stabilization of the Jahn–Teller distortion by hydrogenation of one of the carbon atoms at the edge of a mono-vacancy has been elucidated. This work is the first analysis of the behavior of a graphene vacancy at room temperature in contact with water based on a quantum mechanical molecular dynamics method, where both graphene and solvent are treated at the same level.

## 1. Introduction

“Graphene” has been a hot topic during recent decades. The expectations from the wonder 2D-material had been very high. However, it was soon realized that the magic properties of the idealized, isolated, flat carbon monolayer are strongly affected in real systems.

Already in the thirties of the last century, it was demonstrated that even thermal fluctuations should destroy long-range order in two-dimensional lattices at any finite temperature, resulting in their melting. A clear explanation can be found in reference [1]. Therefore, at room temperature a layer of graphene becomes corrugated. Experimentally, it is not easy to obtain the amplitude of the corrugation. STM analysis provides an excellent atomic resolution on the plane but is less precise in the vertical direction. Furthermore, the interaction of the tip with the surface can induce additional corrugation. Anyway, there is plenty of experimental evidence that characterizes the corrugation. One interesting procedure is the analysis of nano-beam electron diffraction patterns as a function of the incidence angle. These studies of transmission electron microscopy (TEM) have revealed that suspended graphene sheets show elastic random deformations involving all three dimensions with variations in the surface normal by about 1 nm [2,3,4]. This corrugation appears almost static since the experiment takes a long time. In contrast, fast and small thermal oscillations are difficult to observe experimentally.

The second issue is that for practical purposes, graphene needs to be supported. According to the strength of the interaction with the substrate, additional perturbations of its properties can be induced. In this case, corrugation can also appear, although of a different nature than the one mentioned above, resulting in Moiré patterns due to the mismatch of the graphene and substrate lattice constants [5,6,7]. Also, Moiré patterns can appear on HOPG, since the topmost layer may be shifted or rotated by mechanical or chemical means [8,9,10,11].

Next, a variety of defects produced during preparation processes are unavoidable. We can distinguish between intrinsic, such as the Stone–Wales defect, created by rotation of one single pair of carbon atoms, single and multiple vacancies, where carbon atoms of the hexagon ring are missing, line defects, caused at grain boundaries of different crystallographic orientations, and extrinsic defects, originated by foreign adatoms or substitutional impurities [12,13]. In many cases the introduction of these defects can be carried out systematically, in order to tune optical, electronic, magnetic, or catalytic properties [14,15].

Finally, a large number of applications, such as water desalination and energy storing, require the presence of a solvent [16,17,18]. The modeling of solid/liquid interfaces is a challenge that requires much effort in order to provide fundamental insights. Although the carbon surface is hydrophobic, it is also well documented that water plays an important role in many processes that take place at this interface [18].

Concerning the theoretical framework, DFT calculations are widely used to predict structural changes, energetics, and electronic and magnetic properties. These are valuable attempts that have shed light on the understanding of many aspects. However, these results are valid at 0K; therefore, thermal fluctuations effects are missing. A powerful method to investigate the dynamics is classical molecular dynamics (CMD). This is especially efficient to describe the behavior of large ensembles of solvent molecules. However, the lack of reliability of the force field to describe quantum chemically the carbon side of the interface and its interactions with the solvent molecules inhibits the proper description of the whole system.

Cicero et al. [19] investigated water confined in nanotubes and between graphene sheets, and Li et al. [20] investigated water nanodroplets in contact with graphene and monolayer BN by quantum mechanical molecular dynamics (QMMD). However, both works used flat graphene layers, which remain frozen during the simulations.

Interestingly, the development of the self-consistent charge density functional tight-binding method and its combination with molecular dynamics allowed treating both sides of the interface at the same level. This approach has been successfully applied to study the diffusion of vacancies and the annihilation of dislocation pairs in graphene at high temperatures [21,22].

In the present contribution, we shall apply this approach focusing on the dynamics of a mono-vacancy at the top layer of a graphene bilayer. Also, we shall extend this approach to investigate the interface resulting from the contact of graphene with water, allowing the movement of all atoms, including the carbons of the graphene layers.

On one hand, DFT calculations at 0K describe very well the static stable structures. On the other hand, experimental work for structural characterization such as STM and HR-TEM provide images resulting from time periods of the exposures on the order of seconds. Therefore, they represent average configurations. Another limitation of these experimental techniques is the perturbations introduced by the STM-tip and by irradiation with energetic particles, such as electrons or ions. Therefore, our results from QMMD calculations in the framework of DFTB+ can shed some light on details of the dynamics of fast processes occurring in short time scales. This work is the first analysis of the behavior of a graphene vacancy at room temperature in contact with water based on a quantum mechanical molecular dynamics method, where both graphene and solvent are treated at the same level. This also complements a previous publication about desorption of hydrogen from graphene induced by charge injection [23].

## 2. Methods

We performed our calculations using the package DFTB+ from Bremen [24,25], which is based on a tight binding model. The interactions are parametrized through the Slater–Koster files. In particular, we use the 3ob-3-1 set [25], which provides graphene–water interactions of good quality. Dispersion corrections based on the Grimme schema [26] were added within the DFT-D3 option. Our unit cell (25.7 Å × 19.8 Å × 15.985 Å) contains a bilayer of graphene, with one vacancy in the top layer (383 C atoms) and 132 water molecules. All atoms were treated at the same level. Cyclic boundary conditions were applied in all directions, and spin polarization was also included. A NVT ensemble was considered for the molecular dynamic simulations using the velocity Verlet algorithm. All atoms were allowed to move during the simulation. The temperature was kept at 300 K using a Berendsen thermostat. The Brillouin zone was sampled at the Γ point, but tests were performed using 4 × 4 k-points. The results show similar accuracies for our purposes. Typical runs were of the order of 20–30 ps with a time step of 0.5 fs.

## 3. Results and Discussion

### 3.1. Structural Analysis

Figure 1 shows the starting conditions of the graphene bilayer with one vacancy in the top layer, and their first neighbor atoms at the border (labeled as A, B, and C), the second neighbors (labeled as D, E, F, G, H, and I), and the third neighbors (labeled J, K, and L). The original flat surface corrugates immediately after starting the MD simulation, and these structural deformations are dynamical. Figure 2 shows snapshots from two different perspectives of the bilayer graphene with one mono-vacancy in top in contact with water. The corresponding movies in the absence and in the presence of water are available in the Appendix A. As mentioned in the introduction, this behavior is not surprising since thermal fluctuations should destroy long-range order, resulting in melting of the 2D lattice at any finite temperature [1]. The interaction between bending and stretching long-wavelengths phonons could stabilize atomically thin membranes through their deformation in the third dimension [27]. We have also observed corrugation in the case of a graphene bilayer without defects [23]. In our simulations, which allow moving all the atoms, this effect results naturally from the thermal fluctuations, since DFTB treats all atoms at the same level. In most classical molecular dynamic simulations of graphene in contact with water, the corrugation is introduced artificially by means of algorithms applying random or periodical distortions [28,29,30,31] as starting conditions. Later, during the runs, the graphene structure is kept frozen. On the contrary, our simulations are more realistic since all the atoms are allowed to move at the same time.

Figure 3 shows the oscillations of the amplitude out-of-plane of the z coordinates of all carbon atoms, averaged over each graphene layer for both top and bottom layers, ($\tilde{z}={\left[n(Top/Bottom)\right]}^{-1}\sum_{i}^{n(Top/Bottom)}z_{i}$). The standard deviation is actually small (0.015 Å), but the fluctuations of the individual carbon atoms are an order of magnitude larger (about 0.18 Å). This averaged amplitude of the distortions out-of-plane of the z positions of the carbon atoms oscillates regularly and fast in the absence of water with a frequency of about 2.6 × 10^12^ Hz. In the presence of water, the amplitude of these oscillations is larger and random. It is no more possible to identify any characteristic frequency (see blue and cyan curves in Figure 3). The mean separation between the two layers is about 3.3 Å, and it does not change significantly by the presence of water.

An interesting result is the observation of the dynamical structural reconstruction of the graphene mono-vacancy into a Jahn–Teller defect [32]. This symmetry breaking has been predicted from early DFT calculations [33]. It occurs in order to lower the energy (decrease by about 0.2 eV). This reconstruction involves the formation of a closed five-fold ring resulting in a weak bond between two carbons of the border of the vacancy, while the opposite unsaturated carbon is displaced out of the plane. According to the calculations of Popović et al. [32], a large delocalization of the carbon nuclear wave functions around the vacancy occurs, leading to a significant broadening of the Jahn–Teller active sp^2^σ electron states. Therefore, they concluded that the vacancy forms a dynamic Jahn–Teller center.

Experiments using aberration-corrected transmission electron microscopy (AC-TEM) in combination with the use of a mono-chromated electron beam [34] have identified two distinct configurations of the mono-vacancy, the reconstructed Jahn–Teller asymmetric mono-vacancy (r-MV) and a symmetric mono-vacancy (s-MV), both stable over extended exposures [35]. The authors suggest, in accord with the theoretical work of El-Barbary et al. [33] and Popović et al. [32] that one possibility to the observation of the s-MV is that the image is a time averaged superposition of the three degenerated states, which dynamically switch between them due to thermal fluctuations. The estimated barrier to pass from one state to the other has been estimated to be of the order of 0.1 eV [33].

Our simulations confirmed this idea, since we can follow the process in a short timescale not accessible experimentally. Figure 4 shows the results of typical runs for the distances between the unsaturated carbon atoms at the border of the vacancy (A, B, and C) for a “clean” vacancy (upper panel) and for when one of the nearest carbon atoms (A) is hydrogenated (bottom panel). First, we shall analyze the “clean” vacancy (Figure 4, upper panel). The distance between the pairs of neighbor’s atoms alternates between 2.25 Å and 2.55 Å. The shorter distance corresponds to a weak bond between the carbon atoms closing one pentagon. The switching between these alternating bonds is very fast and random. These results agree with the estimated averaged displacement of the nearest carbon atoms to the vacancy from [32]. When one of the carbons at the border of the vacancy (in this case A) is saturated by a hydrogen atom (bottom panel), the carbon atoms (B and C) are more strongly bonded than in the first case, as evidenced by the shorter distance between them (black line). The formation of a pentagon can be observed during the whole simulation (see movies in the Appendix A). There are no more alternating bonds between these two carbon atoms B–C and the carbon A, which now is displaced into the vertical direction at a farther distance (red and green lines).

The presence of water does not have any significant effect. A better representation is given by the bond distance distribution shown in Figure 5 (upper panel). The results for the distribution of distances between first, second, and third neighbor’s carbon atoms are shown in comparison with the corresponding distances in pristine graphene. The bonds between the closest carbon atoms are similar to those of pristine graphene (A–D, A–E; B–I, B–H; C–F, C–G; mean value of about 1.43–1.44 Å). The distances between the second neighbors (D–E, E–F, F–G, G–H, H–I, I–D) are slightly shorter than in pristine graphene (mean values of 2.43 Å and 2.47 Å, respectively), indicating a weak compression of the rings. The distances between the farther atoms’ pairs (J–K, J–K, K–L) do not change significantly in comparison with pristine graphene. They only show a broader distribution and small asymmetries at the lower values, also an indication of slight compression. These results are also in agreement with the calculations in [32]. The analysis of the distances between the atoms at the border of the vacancy (A, B, and C) shows the most interesting features. There is a split in the distance distribution around the mean value observed for pristine graphene (mean value 2.47 Å) that reflects the alternating formation of the Jahn–Teller distortion between these atoms. Nevertheless, the mean value is about 2.41 Å, similar to the distance by the s-MV (2.3–2.4 Å) estimated from experiments [35]. This representation is a complementary view of Figure 4 (upper panel) and evidences the oscillating formation of the pentagon between the three atoms of the vacancy’s border.

Concerning the stable reconstructed vacancy (r-MV) observed experimentally in [35], the authors mentioned the possibility that this configuration could be stabilized by the functionalization of the under-coordinated carbon radical by a low mass contaminant, such as hydrogen, precluding any oscillation of the pentagon by bond switching between the edge atoms (A, B, and C). In order to investigate this possibility, we have carried out simulations hydrogenating the unsaturated carbon atom A. This process induces a farther displacement of atom A perpendicular to the surface (from 3.52 Å to 4.11 Å) and the concomitant farther stretching out of plane of the two nearest carbon atoms D and E (from 3.44 Å to 3.77 Å). But the most important effect is the stabilization of the Jahn–Teller distortion by the B–C bonding and the disappearance of alternating switch with atom A. Figure 4 (bottom panel) shows that practically during the whole run of 30 ps this bond remains stable. The corresponding distance distributions are shown in Figure 5 (bottom panel). The split is now more pronounced (mean values of 2.12 Å and 2.74 Å) in comparison with the free vacancy (peaks at 2.25 Å and 2.55 Å). Robertson et al. [35] performed a careful geometric phase analysis and bond length measurements for the r-MV and obtained values of 1.9 Å and 2.8 Å for these distances. They also estimated the distances between A–D and A–E carbon nearest atoms and found bond compression to 1.37 Å and 1.21 Å, respectively, in comparison with the bulk nearest C-C bond. Our mean value is about 1.40Å, also slightly lower than the corresponding value for pristine graphene (1.44 Å) but somewhat larger than the experimental value. However, the authors of the experimental work [35] noticed that the nature of AC-TEM imaging physically precludes the measurements of any bond length contribution in the *z*-axis; thus, any out-of-plane displacement of the under coordinated carbon atom would lead to an illusory bond compression in the TEM image, due to the corresponding projection of the bond into the 2Dx-y plane. Therefore, we also calculated this distance projected into the plane x-y and obtain effectively a lower mean distance of 1.16 Å.

In the supporting information are movies available comparing both systems (Appendix A), with and without functionalization.

In all our simulations, we never detected the diffusion of the vacancy along the surface. This phenomenon has been observed on simulations at higher temperatures (2000–3000 K). In our case, it is not surprising since the activation barrier for this process has been estimated to be larger than 1.7 eV [33]. Therefore, the diffusion at room temperature is improbable.

### 3.2. Charge Analysis

An interesting additional advantage of DFTB+ is that the charge distribution between the atoms can be easily obtained from the orbital occupation analysis. In this section, we shall discuss the atomic charge distribution for the different investigated systems.

Figure 6 (upper panel) shows the charge distribution between the top layer (with one vacancy) and the bottom layer of graphene in the absence of water. There is a slight polarization between both layers due to the vacancy in the top layer. At the bottom panel of Figure 6, the corresponding charges are shown in the presence of water. Here, in addition to the polarization between both graphene layers, a stronger polarization appears due to the presence of water; the negative charge on water (mean value: −0.1 |q_e−_|) is compensated by the total charge on the graphene bilayers (mean value: +0.1 |q_e−_|). Li et al. [20] also found for the droplet–graphene system by integration of the electronic density of DFT calculations a positive charge on the contacting surface of graphene, although somewhat larger (+1.3 |q_e−_|) than ours. Similar to our results for the out of plane coordinates fluctuations, the charge variations in the presence of water are larger than in its absence.

Figure 7 shows another view of the charge distribution. Here, the broadening of the distributions in the presence of water becomes evident.

It is also interesting to analyze the charge on the neighboring atoms of the vacancy separately. Figure 8 (upper panel) shows the charge distribution on the atoms at the edge of the vacancy (A + B + C), on the next neighbor to the border atoms (D + E + F + G + H + I), and on the farther third neighbor atoms (J + K + L). The borders of the vacancy are negatively charged, the mean value in the absence of water being slightly less negative (−0.28 and −0.26 |q_e-_|, respectively). The charge distribution in the presence of water is broader than in its absence. This charge is practically compensated by the positive charge (+0.25 |q_e−_|) on the next neighbor (D + E + F + G + H + I), see Figure 8 (bottom panel). Therefore, we can conclude that this charge perturbation is almost a local effect.

A detailed analysis of the charge distribution on the individual atoms at the border of the vacancy (A, B, and C) (Figure 9) reveals the effect of the alternating Jahn–Teller distortion. Similar to the effect observed by the distribution of carbon bonds, the charge distribution splits into two regions. When two carbon atoms form the weak bond closing a pentagon, the third one is strongly negatively charged, while the two forming the bond are less charged. In the case that the carbon atom (A) is hydrogenated, this atom shows negative charge (mean value: −0.12 |q_e−_|), and the others two forming the weak bond of the distortion are also negatively charged (mean value: −0.1 |q_e−_|), while the hydrogen bonded to A is positively charged, practically compensating the charge on A (mean value: +0.1 |q_e−_|).

Finally, two views (into zx and zy planes) of the superposition of the coordinates of all atoms of one typical run and the corresponding charges on the carbon atoms are shown in Figure 10. The water molecules are orientated mainly pointing at least one hydrogen to the graphene layer, as can be established from the red regions closest to graphene. The fluctuations of position of the carbon atoms around the vacancy are larger, and therefore they become closer to water (upper panel). The region around the vacancy also shows the oscillation on the charge of the neighbors’ carbon atoms (bottom panel). In the supporting information are shown movies comparing the bilayer of graphene in contact with water with and without the mono-vacancy in the top layer (Appendix A). An artificial C–H bond length of 2.8 Å was imposed in order to visualize better the interaction of water with the graphene surface. The water molecules approach the surface to 2.8 Å or least “walk” some steps at short distances before reorienting and moving apart. It is noticeable that the water molecules that approach the vacancy remain there longer.

In the analysis of these results in order to obtain the atomic density, the electrical field and the electrostatic potential across the interfaces are shown in Figure 11. The upper panel displays a profile of the normalized atomic density in the z-direction averaged in the parallel planes xy and normalized to the total number of each particular atom. The hydrophobicity of graphite and multilayers of graphene is well known. Although recently the wettability of graphene has been under intense debate [36], there are plenty of works that report the hydrophobicity of graphene. It seems that the substrate on which graphene is supported, and the number of layers play an important role. Therefore, it is not surprising that water molecules are excluded from a wide region at the interface. There is an interesting early work of Nagy [37], where the structure of water at graphite is investigated by STM. He estimated the tunnelling barrier height and found it to have a maximum at about 0.2–0.25 nm from the surface. He concluded that it is related to the positions of water molecule oxygens of a distinct water layer next to the surface, whose molecules are mobile. This exclusion has been previously observed both by CMD [38,39] and by QMMD [19]. They found the thickness of this exclusion region to be about 2.5 Å. However, in their simulations the graphene layer is flat and remains fixed. In our case, (see zoom view in in the inset of the upper panel of Figure 11), the actually zero density encloses about 1 Å. If we take as reference the mean value of z of the corrugated graphene, the distance to the first maximum for oxygen density profile is about 3.2 Å, similar to the results of Ho and Striolo [38,39]. The density of water seems to be larger for the layer closest to the surface, similar to previous results [23]. The bottom panel of Figure 11 shows the resulting electric field obtained by integration of the planar charge perpendicular to the surface and the corresponding electrostatic potential obtained by further integration of the field. We should keep in mind that the graphene layer in contact with water becomes positively charged (see Figure 6, bottom panel). Therefore, in the empty region between graphene and water, first the field decreases, and afterward, when the hydrogen atoms of the first layer of water emerge, the field increases again, creating a dipole. At a larger distance, a second dipole becomes evident from the interplay between the oxygen and the hydrogen atoms of the first water layer.

Concerning the electrostatic potential, first it increases negatively into the bulk of the electrolyte until a screening effect of the solvent becomes evident. At the end, since the dipole formed with the bottom layer of graphene is larger, a net dipole potential appears due to the asymmetry introduced by the mono-vacancy in the top layer of graphene.

## 4. Conclusions

We have performed quantum mechanical molecular dynamics in the framework of DFTB+ for a bilayer of graphene containing one mono-vacancy in the absence and in the presence of water, treating all components of the system at the same level. We have obtained a clear view of the dynamics of the systems on a timescale inaccessible experimentally. We summarize below the most relevant results.

The graphene bilayers corrugate at room temperature as a consequence of thermal fluctuations. The twists out of plane show a characteristic high frequency of about 2.5 × 10^12^ Hz for the bilayer in vacuum. Therefore, these distortions should be very difficult to be observed experimentally. The presence of water in contact with the bilayer increases the amplitude of the oscillations in the z-direction, and they are not more periodic. There is a polarization between graphene and water, with the graphene bilayer showing a positive charge compensated by the negative charge on water. At the top layer of graphene, the carbon atoms nearest to the mono-vacancy are negatively charged, while the next nearest neighbors are positively charged.

An interesting result is the dynamics of the Jahn–Teller distortion. In the case of the “clean” vacancy, two of the three carbon atoms at the edge occasionally bond to each other. These switching between the bonds alternate randomly. When one of the carbon atoms with dangling bond is saturated with one hydrogen atom, the Jahn–Teller distortion between the others two carbon atoms is stabilized. These results explain quite well the previous experimental finding [35].

## Figures and Tables

**Figure 1 nanomaterials-13-02038-f001:**
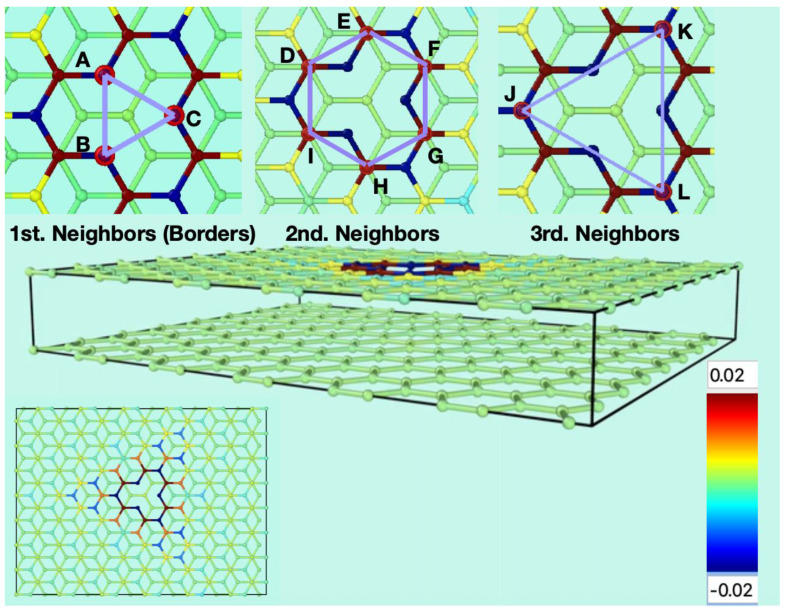
Starting conditions of the simulations for the flat bilayer of graphene containing one mono vacancy in the top layer (perspective and top views). The three insets on the upper part show the carbon atoms surrounding the vacancy. A, B and C are the carbon atoms containing the σ dangling bonds. D, E, F, G, H, and I are the next carbon atoms to A, B and C. J, K and L are the third neighbours to the vacancy. The colour gradient shows the charge scale range for the atoms.

**Figure 2 nanomaterials-13-02038-f002:**
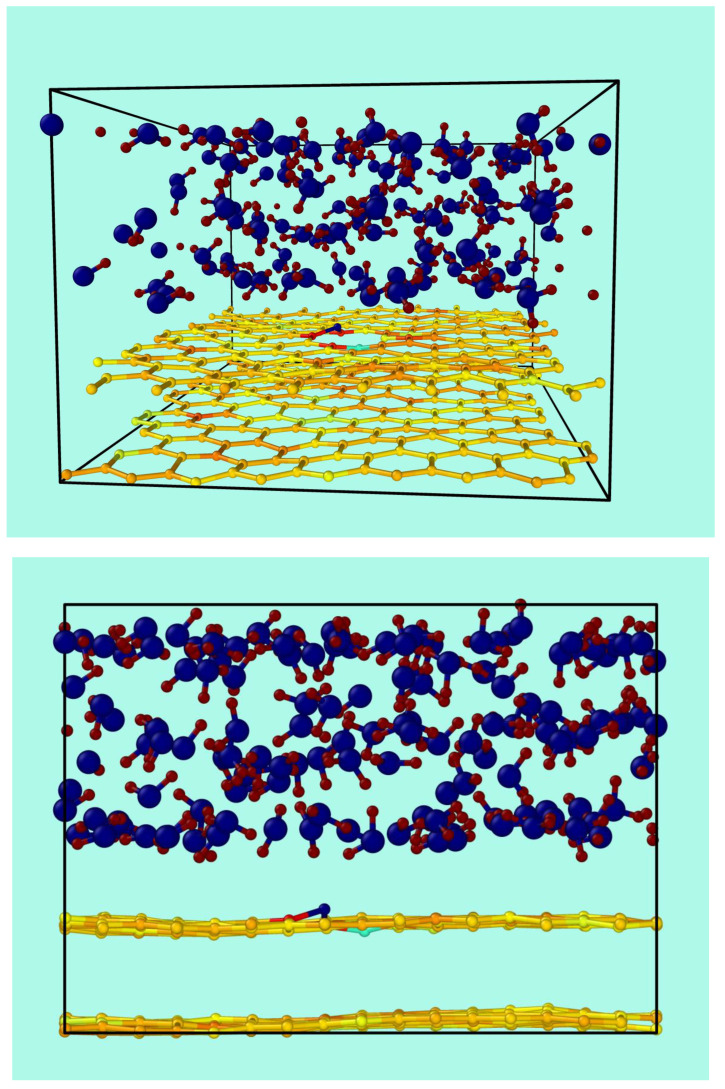
Two views of a snapshot for a typical simulation run of the bilayer of graphene containing a vacancy in contact with water.

**Figure 3 nanomaterials-13-02038-f003:**
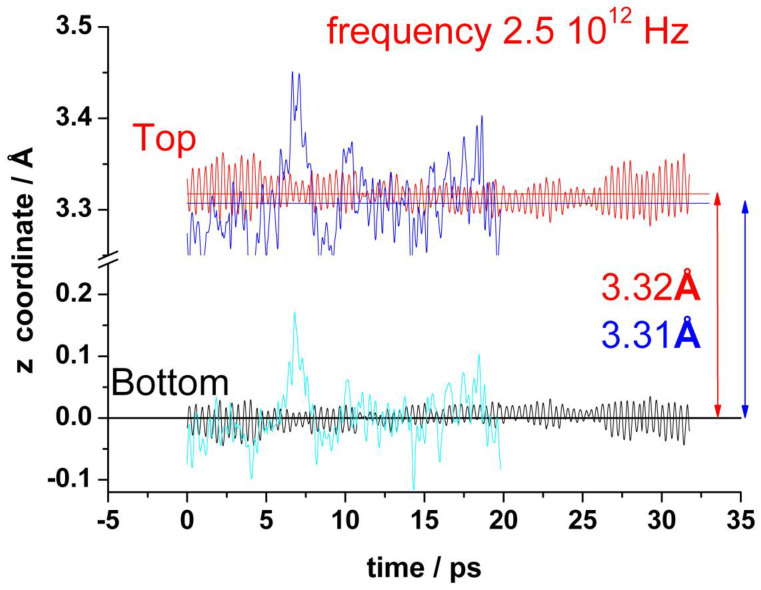
Variation of the average z-coordinate for top and bottom layers of graphene during a typical run. In the absence of water: red (top) and black (bottom). In the presence of water: blue (top) and cyan (bottom).

**Figure 4 nanomaterials-13-02038-f004:**
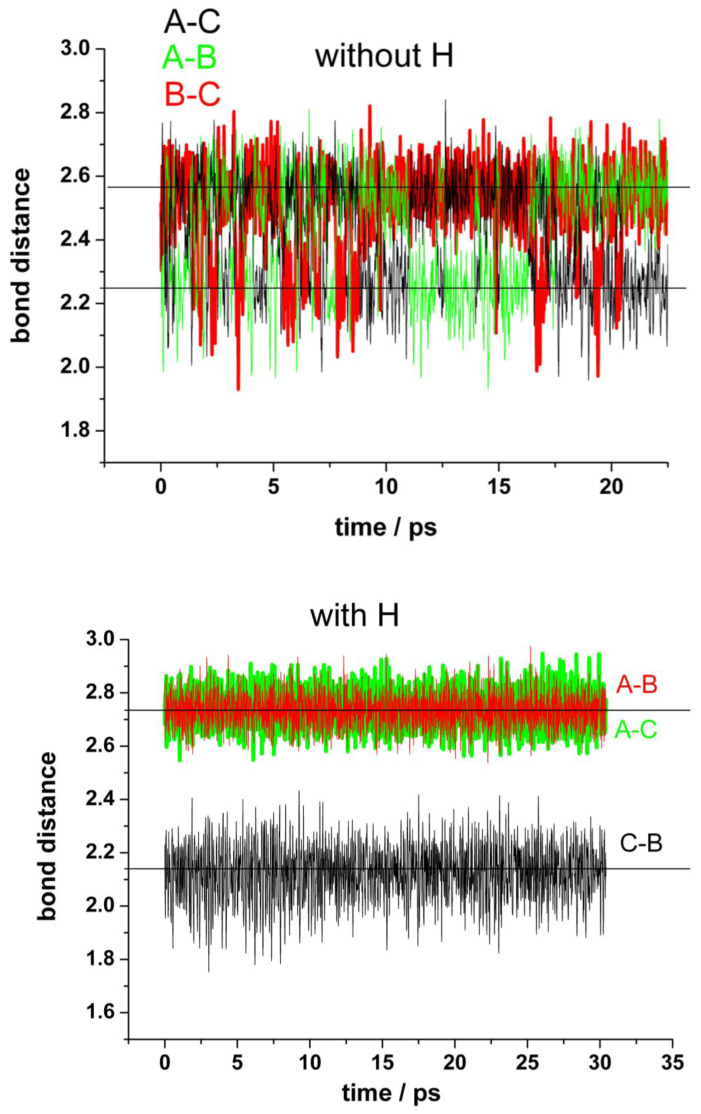
Fluctuations in the distances between the carbon atoms (A,B,C) at the edge of the vacancy in the absence of water. **Upper panel**: mono-vacancy (MV) with the three border carbon atoms unsaturated. **Bottom panel**: mono-vacancy with the A carbon atom at the border saturated by one hydrogen atom.

**Figure 5 nanomaterials-13-02038-f005:**
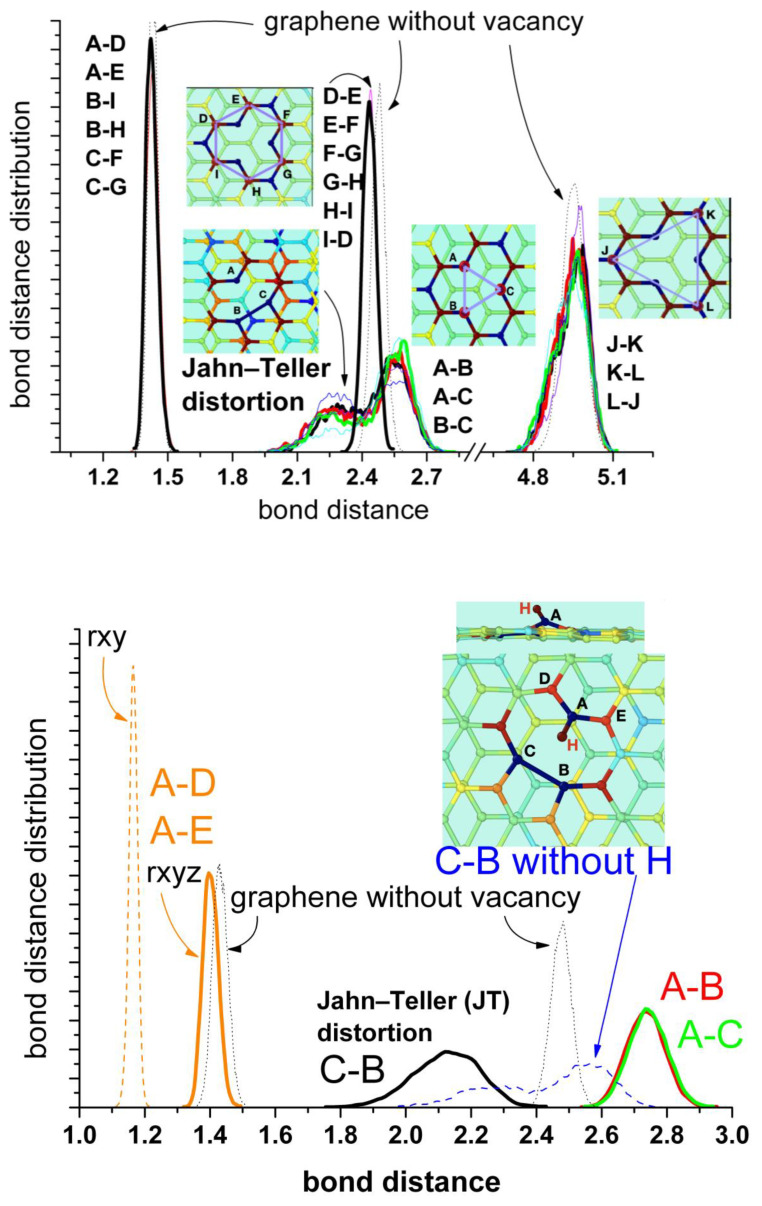
Bond distance distributions between first, second and third neighbour’s carbon atoms. **Upper panel**: mono-vacancy (MV) with the three border carbon atoms unsaturated. **Bottom panel**: mono-vacancy with the A carbon atom at the border saturated by one hydrogen atom. For comparison the blue dashed curve at bottom corresponds to the B-C distance for the first case without hydrogen shown in the upper panel. The thin lines correspond to the results in the absence of water, while the thick lines in the presence of water. The dotted lines correspond to the equivalent bonds in pristine graphene. The insets show the geometric configuration of the mono-vacancy.

**Figure 6 nanomaterials-13-02038-f006:**
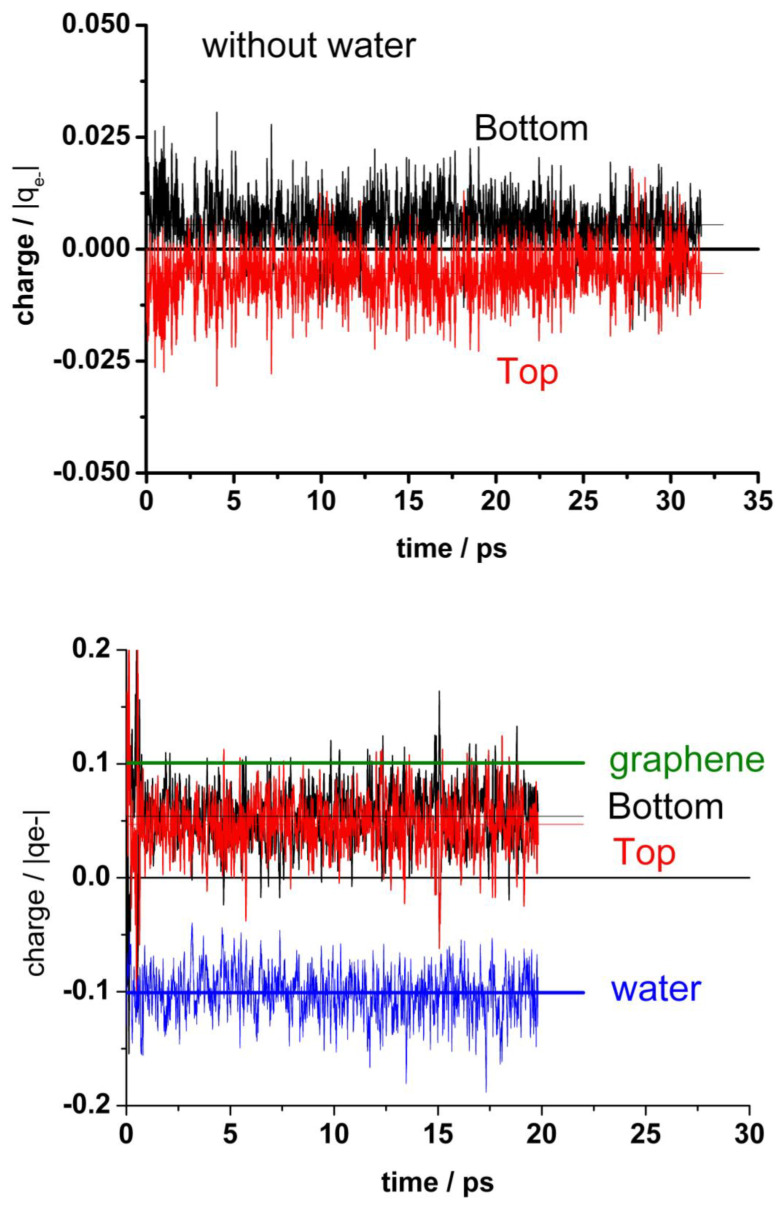
Charge allocation in the different components of the systems. **Upper panel**: Graphene bilayer with the mono-vacancy (MV) in the top layer in the absence of water. **Bottom panel**: Graphene bilayer with the MV in the top layer in the presence of water. The dotted lines indicate the mean value for bottom and top graphene layers. The black thick line at upper panel and the olive line at bottom panel correspond to the mean value for the whole graphene (**bottom** + **top**). The blue line at bottom indicates the mean value for the whole water.

**Figure 7 nanomaterials-13-02038-f007:**
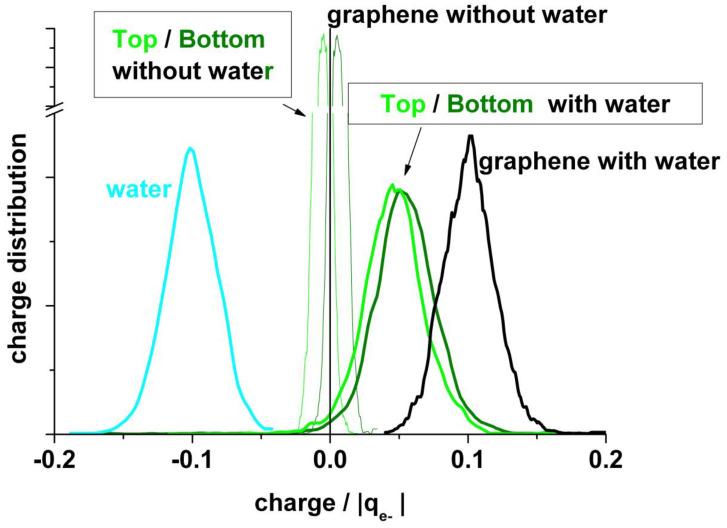
Comparison of statistical charge distributions from typical runs like in Figure 6 between systems in absence and in presence of water.

**Figure 8 nanomaterials-13-02038-f008:**
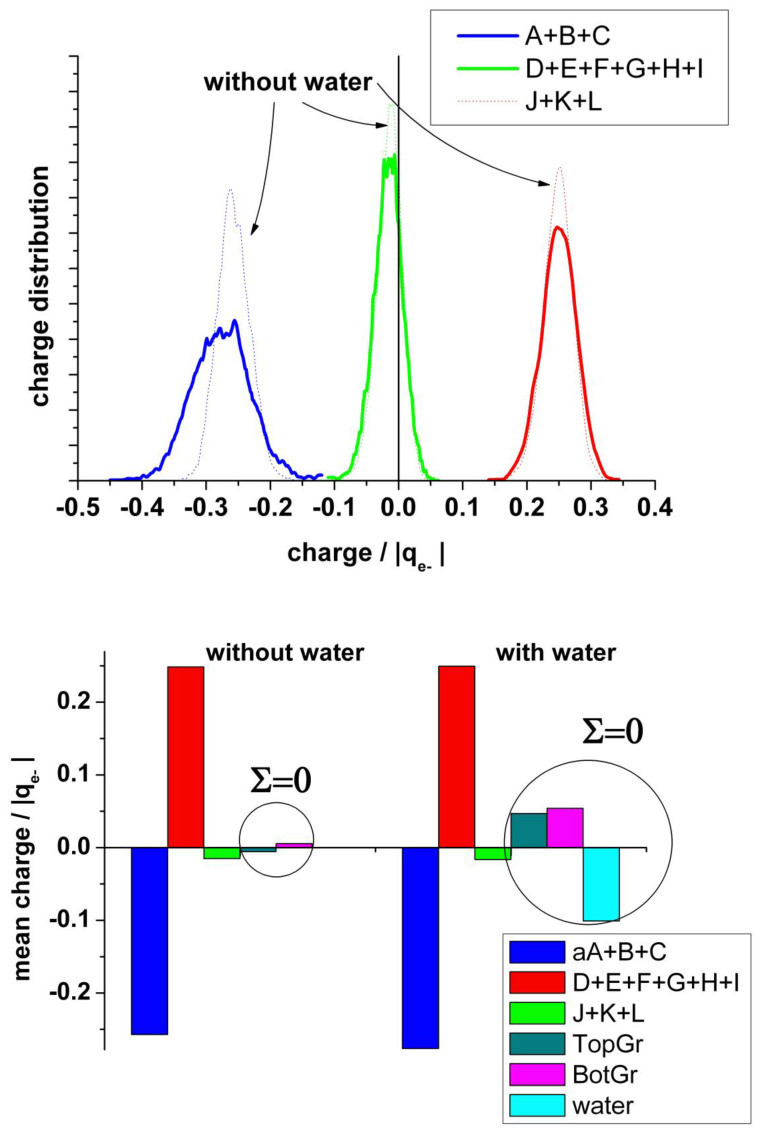
**Upper panel**: Statistical charge distribution of the atoms surrounding the vacancy in the presence (solid curves) and in the absence (dotted curves) of water. Blue curves: total first neighbours A + B + C; green curves: total second neighbours D + E + F + G + H + I; red curves: total third neighbours J+K+L. **Bottom panel**: Comparison of the mean values from upper panel, with and without water.

**Figure 9 nanomaterials-13-02038-f009:**
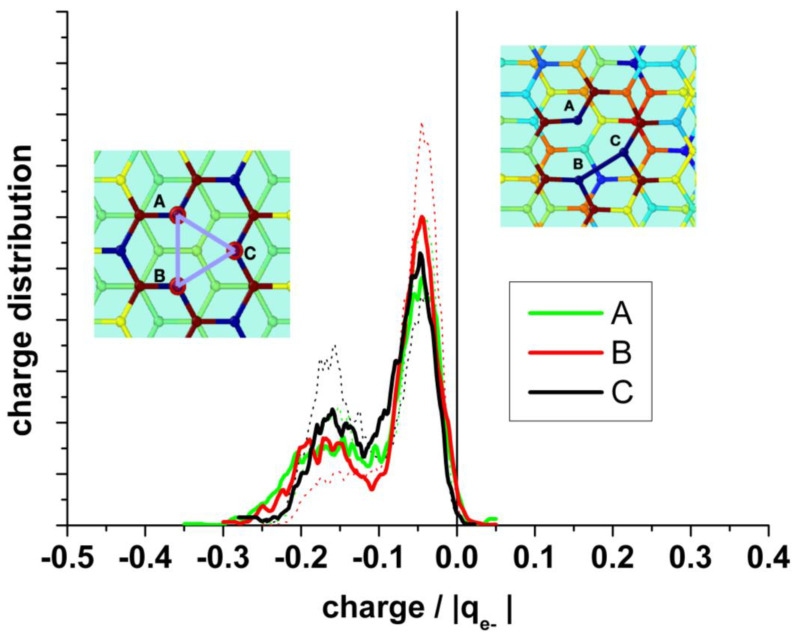
**Upper panel**: Statistical charge distribution of the individual atoms surrounding the vacancy in the presence (solid curves) and in the absence (dotted curves) of water. **Bottom panel**: Statistical charge distribution of the individual atoms surrounding the vacancy when the carbon atom A is hydrogenated.

**Figure 10 nanomaterials-13-02038-f010:**
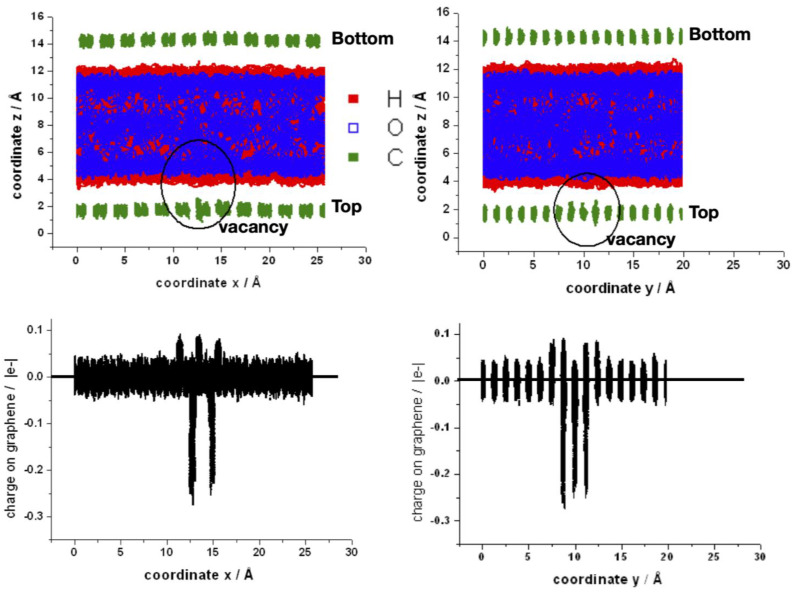
**Upper panel**: space distribution of the superposition of z-coordinates for a typical MD-simulation of the bilayer with a mono-vacancy in the top layer along the x- (**left** side) and y- (**right** side) coordinates. The circle indicates the position of the mono-vacancy. **Bottom panel**: space distribution of the superposition of individual atomic charges of carbon atoms along the x- (**left** side) and y- (**right** side) coordinates.

**Figure 11 nanomaterials-13-02038-f011:**
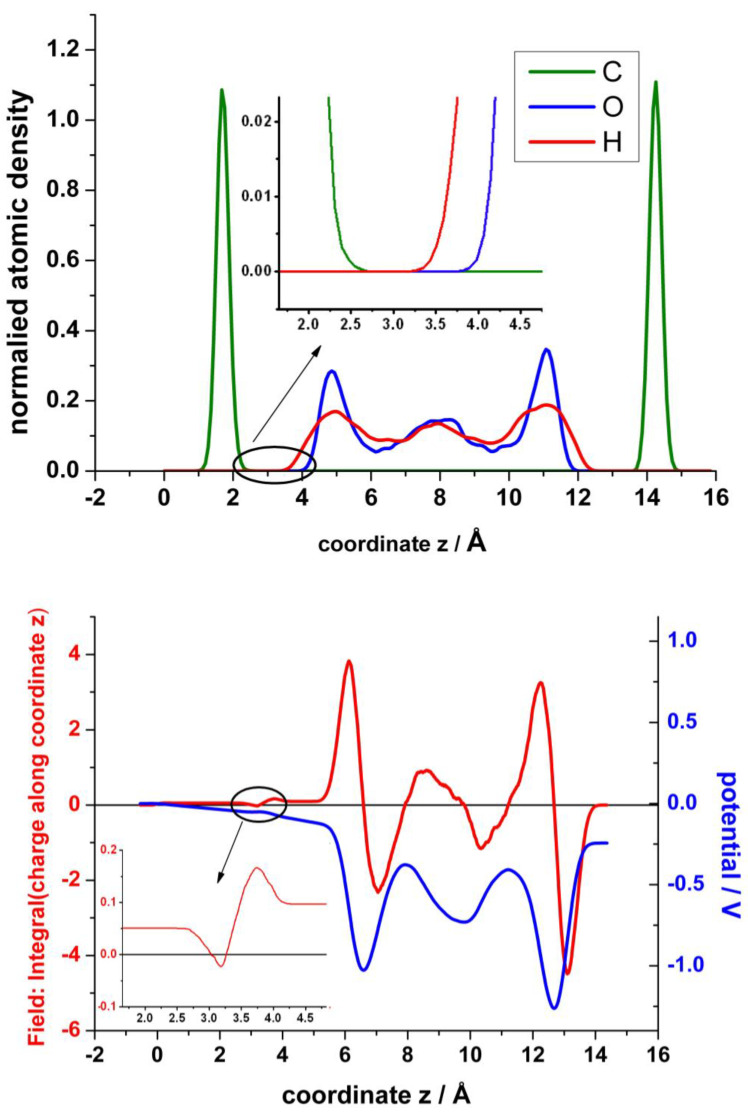
**Upper panel**: Normalized particle distribution in the perpendicular direction of the graphene surface. **Bottom panel**: Distribution of the field, obtained by integration of the charge (red), and of the electrostatic potential (blue), obtained by further integration of the field. The insets show a zoom view of the densitiy and of the field in the region between the top layer of graphene and the first layer of water.

## Data Availability

Data Availability on request.

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
