# Peer review of "Structural Dynamics in the Presence of Water of Graphene Bilayers with Defects"

_nanomaterials, 2023, doi:10.3390/nano13142038_

Round 1
Reviewer 1 Report
The study presented here dealt with the detailed structural description of a mono-vacancy in a graphene bilayer based on DFT and MD simulations. The published results are in good agreement with previously published experimental results, which support the stabilization of the Jahn-Teller distortion by the hydrogen atom of a carbon atom in an edge position. The whole manuscript is well structured and logical and deserves to be published. However, the quality of the figures is really poor, the resolution of the figures is terrible. This needs to be improved before publication.
Considering the above-mentioned, I suggest this manuscript for publication in an MDPI journal “Nanomaterials” after minor revision.
—
Author Response
I thank the reviewer for the evaluation of my work. Now, I have improved the
manuscript according to their suggestions, and I trust that the article is ready for publication.
Yes! I agree that the resolution of the figures is very poor... I have now separately
included the figures with a better quality.
Reviewer 2 Report
In this paper, the dynamics of a bilayer of graphene containing one mono-vacancy in the top layer has been investigated in the framework of DFTB in the absence and in the presence of water. The presence of water enhances the displacement of carbon atoms in the perpendicular direction to the surface. They results explain very well a variety of experimental findings. In particular, the stabilization of the Jahn-Teller distortion by hydrogenation of one of the carbon atoms at the edge of a mono-vacancy has been elucidated. However, there are some issues that need to be clarified. The detailed comments are as follows:
1. The advantages of this study can be described in introduction. Finally, a large number of applications, such as water desalination and energy storing, require the presence of a solvent. Some references should be added. Separation and Purification Technology, 307 (2023) 122716; Surfaces and Interfaces, 36 (2023) 102564;
2. What is the innovation point of this paper?
3. Please improve the clarity of the picture.
4. The names in Figure 4 are not shown completely.
5. Can you only study structures at 0 K?
In this paper, the dynamics of a bilayer of graphene containing one mono-vacancy in the top layer has been investigated in the framework of DFTB in the absence and in the presence of water. The presence of water enhances the displacement of carbon atoms in the perpendicular direction to the surface. They results explain very well a variety of experimental findings. In particular, the stabilization of the Jahn-Teller distortion by hydrogenation of one of the carbon atoms at the edge of a mono-vacancy has been elucidated. However, there are some issues that need to be clarified. The detailed comments are as follows:
1. The advantages of this study can be described in introduction. Finally, a large number of applications, such as water desalination and energy storing, require the presence of a solvent. Some references should be added. Separation and Purification Technology, 307 (2023) 122716; Surfaces and Interfaces, 36 (2023) 102564;
2. What is the innovation point of this paper?
3. Please improve the clarity of the picture.
4. The names in Figure 4 are not shown completely.
5. Can you only study structures at 0 K?
Author Response
I thank the reviewer for the evaluation of my work. Now, I have improved the
manuscript according to their suggestions, and I trust that the article is ready for publication.
1. The advantages of this study can be described in introduction. Finally, a large
number of applications, such as water desalination and energy storing, require the presence of a solvent. Some references should be added. Separation and Purification Technology, 307 (2023) 122716; Surfaces and Interfaces, 36 (2023) 102564;
I have added the two suggested references ([17] and [18]).
2. What is the innovation point of this paper?
I have added in the abstract and the introduction:
This work is the first analysis of the behaviour of a graphene vacancy at room
temperature in contact with water based on a quantum mechanical molecular
dynamics method, where both graphene and solvent are treated at the same level.
3. Please improve the clarity of the picture.
Yes! I agree that the resolution of the figures is very poor... I have now separately included the figures with a better quality.
4. The names in Figure 4 are not shown completely.
I think, the reviewer means that the right side of the Figure 4 was not explained in the text. I have added:
When one of the carbons at the border of the vacancy (in this case A) is saturated by a hydrogen atom (right side), the carbon atoms (B and C) are more strongly bonded than in the first case, as evidenced by the shorter distance between them (black line). The formation of a pentagon can be observed during the whole simulation (see movies in the SI). There are no more alternating bonds between these two carbon atoms B - C and the carbon A, which now is displaced into the vertical direction at a farther distance (red and green lines).
5. Can you only study structures at 0 K?
The whole calculations presented here were performed at 300K, since the goal of this investigation is to understand the dynamical behaviour of the system at room temperature. Of course, using DFTB the structures at 0K can also be studied, but such results are not relevant for this work.
Round 2
Reviewer 2 Report
accepted